# Prognostic Potential of Apoptosis-Related Biomarker Expression in Triple-Negative Breast Cancers

**DOI:** 10.3390/ijms26157227

**Published:** 2025-07-25

**Authors:** Miklós Török, Ágnes Nagy, Gábor Cserni, Zsófia Karancsi, Barbara Gregus, Dóra Hanna Nagy, Péter Árkosy, Ilona Kovács, Gabor Méhes, Tibor Krenács

**Affiliations:** 1Department of Pathology, University of Debrecen Clinical Center, Kenézy Gyula Campus, 4031 Debrecen, Hungary; dr.torok.miklos@med.unideb.hu (M.T.); ilona.kovacs@med.unideb.hu (I.K.); 2Department of Pathology and Experimental Medicine, Semmelweis University, 1085 Budapest, Hungary; nagy.agnes@semmelweis.hu (Á.N.); nagy.d.hanna@gmail.com (D.H.N.); 3Department of Pathology, Bács-Kiskun County Teaching Hospital, 6000 Kecskemet, Hungary; csernig@kmk.hu; 4Department of Pathology, Forensic and Insurance Medicine, Semmelweis University, 1085 Budapest, Hungarygregus.barbara@semmelweis.hu (B.G.); 5Department of Oncology, Faculty of Medicine, University of Debrecen, 4032 Debrecen, Hungary; arkosy.peter@med.unideb.hu; 6Department of Pathology, Faculty of Medicine, University of Debrecen, 4032 Debrecen, Hungary; mehes.gabor@med.unideb.hu

**Keywords:** triple-negative breast cancer, apoptosis pathways, apoptosis inducing factor-1, caspase-3, disease prognosis

## Abstract

Of breast cancers, the triple-negative subtype (TNBC) is characterized by aggressive behavior, poor prognosis and limited treatment options due to its high molecular heterogeneity. Since insufficient programmed cell death response is a major hallmark of cancer, here we searched for apoptosis-related biomarkers of prognostic potential in TNBC. The expression of the pro-apoptotic caspase 8, cytochrome c, caspase 3, the anti-apoptotic BCL2 and the caspase-independent mediator, apoptosis-inducing factor-1 (AIF1; gene *AIFM1*) was tested in TNBC both in silico at transcript and protein level using KM-Plotter, and in situ in our clinical TNBC cohort of 103 cases using immunohistochemistry. Expression data were correlated with overall survival (OS), recurrence-free survival (RFS) and distant metastasis-free survival (DMFS). We found that elevated expression of the executioner apoptotic factors AIF1 and caspase 3, and of BCL2, grants significant OS advantage within TNBC, both at the mRNA and protein level, particularly for chemotherapy-treated vs untreated patients. The dominantly cytoplasmic localization of AIF1 and cleaved-caspase 3 proteins in primary TNBC suggests that chemotherapy may recruit them from the cytoplasmic/mitochondrial stocks to contribute to improved patient survival in proportion to their expression. Our results suggest that testing for the expression of AIF1, caspase 3 and BCL2 may identify partly overlapping TNBC subgroups with favorable prognosis, warranting further research into the potential relevance of apoptosis-targeting treatment strategies.

## 1. Introduction

Breast cancer is a significant global health concern, representing the most commonly diagnosed cancer in women worldwide [1]. In 2022, an estimated 2.3 million new cases were diagnosed, and 670,000 deaths were attributed to the disease [2]. The current immunohistochemistry-based molecular classification divides breast cancers into four subtypes—Luminal A-like (hormone receptor-positive/low proliferation), Luminal B-like (hormone receptor-positive/high proliferation), HER2-positive (estrogen receptor negative, *ERBB2*-amplified), and triple-negative (TNBC, based on the absence of estrogen receptor, progesterone receptor and HER2 expression) [3]. TNBC accounts for approximately 15% of all breast cancers [4] with an 8% to 16% lower 5-year survival rate than that of hormone receptor-positive disease [5]. TNBC is a heterogeneous disease, comprising of multiple distinct entities with significant histopathological, transcriptomic and genomic diversity, which complicates its prognostic stratification and therapeutic decision-making [6]. Therefore, searching for biomarkers that may have prognostic potential and offer novel therapeutic targets is highly relevant.

In general, TNBC is characterized by aggressive clinical course, including a higher risk of early recurrence, particularly within the first three years after diagnosis, and propensity for visceral metastases, especially to the lungs, along with frequent spread to the brain [7]. Nevertheless, histopathologically, TNBC may range from low-grade carcinomas with indolent behavior to high-grade neoplasms featured by rapid progression [8]. Salivary gland-like TNBC, including adenoid cystic and secretory carcinomas, with specific fusion genes (*MYB–NFIB*, *ETV6–NTRK3*) and minimal genetic instability, often maintain favorable outcomes [9]. Metaplastic carcinomas of heterologous differentiation show inherent chemoresistance and aggressive clinical course, while the most prevalent form, invasive carcinoma of no special type, reveals remarkable diversity with dense tumor-infiltrating lymphocytes (TILs) associated with improved survival despite high-grade cytology [10]. The molecular classification of TNBC highlights a complex spectrum, which by now consists of four subtypes: basal-like 1 (BL1), basal-like 2 (BL2), mesenchymal (M) and luminal androgen receptor (LAR) [11]. Recent whole-genome sequencing of 198 TNBC cases confirmed LAR, and mesenchymal (MES) subtypes and refined the basal-like category into immunosuppressed (BLIS), and immune-activated (BLIA) subtypes [12]. However, there are still no globally standardized diagnostic criteria for the molecular classification of TNBC.

The TNBC subtypes demonstrate distinct characteristics, prognoses and therapy response. The main therapeutic options of different subtypes include platinum-based chemotherapy and poly-ADP ribose polymerase (PARP) inhibitors for BRCA1 mutant BL1 tumors [13]. The basal/myoepithelial BL2 subtype may potentially respond to receptor tyrosine kinase inhibitor and EGFR-targeted therapies [14]. The MES subtype shows stem-cell-like properties, metaplastic character, moderate chemoresistance and frequently harbors activating mutations in the *PI3K*/*AKT*/*MTOR* axis, suggesting their potential sensitivity to mTOR inhibitors (e.g., everolimus) [15]. The immunomodulatory (IM) subtype with TIL infiltration may react to immune checkpoint (PD-L1, PD-1 and CTLA4) inhibitors, particularly to anti-PD-1/PD-L1 (pembrolizumab, atezolizumab) treatment [16]. The LAR subtype is defined by androgen receptor (AR) overexpression, calcifications and chemoresistance, which may respond to AR antagonists, though with a limited efficiency in monotherapy [17]. However, none of these treatment options proved to show significant outcome benefits.

Apoptosis is the most common form of programmed cell death induced by cell stress, including treatment-related DNA damage [18]. The cascades of events in apoptosis are mediated through partly interrelated intrinsic (mitochondrial) and extrinsic (death receptor) pathways and involve the activation of cysteine-aspartic acid proteases (caspases) (Figure 1). The intrinsic pathway can be induced by cellular stress through the pro-apoptotic BAX (BAK) activation, which can damage mitochondrial membranes and release proteins that are normally involved in cellular respiration, such as cytochrome c (gene *CYCS*) and apoptosis-inducing factor (AIF1 encoded by the *AIFM1* gene). Upon release, CYCS forms apoptosomes with apoptotic protease activating factor-1 (Apaf-1) and procaspase 9, the latter of which is then cleaved into caspase 9 to subsequently activate caspase 3 (CASP3), the primary executioner caspase responsible for DNA fragmentation. When the flavoprotein AIF1 is released, it can directly translocate to the nucleus and may also induce DNA fragmentation and chromatin condensation in a caspase-independent manner. BCL2, an anti-apoptotic protein, plays a critical role in controlling the intrinsic pathway by preventing CYCS release. The extrinsic pathway is activated through the tumor necrosis factor-family cell membrane death receptors (FAS, DR4 or DR5), which signal caspase-8 (CASP8) to induce the cleavage and activation of the pro-apoptotic protein Bid and link the extrinsic into the intrinsic pathway. Subsequent studies have demonstrated that CASP8 can activate CASP3 either directly or indirectly through CYCS release in diverse apoptotic scenarios [18]. In cancer, the apoptotic pathways are often deregulated, leading to uncontrolled cell proliferation and resistance to cell death signals [19]. Thus, apoptosis induction has emerged as a critical mechanism in cancer therapy [20].

In this study, we tested the expression of biomarkers of apoptotic pathways both in silico at mRNA and in situ at protein level and correlated these data with TNBC prognosis. We focused on the major players of caspase-dependent (CASP8, CYCS, CASP3 and BCL2) and caspase-independent (AIF1) pathways (Figure 1). At the protein level, we detected the cleaved form of CASP3 and CASP8, to demonstrate not only the amount but also the activation status of these caspases. Our aim was to identify TNBC patients who could potentially benefit from apoptosis-targeting therapies.

## 2. Results

### 2.1. Apoptosis-Inducing Factor-1 Expression and TNBC Prognosis

A major caspase-independent apoptosis pathway is mediated by the AIF1 protein encoded by the *AIFM1* gene. KM Plotter-based high *AIFM1* transcriptome expression data, gained using the RNAchip method, showed significantly enhanced OS in TNBC when the patient cohort was separated at best cutoff mRNA expression level (logrank *p* = 0.018; HR = 0.48 (0.26–0.9)) (Figure 2A); but it had no significant effect either on RFS or DMFS. Elevated AIF1 protein expression tested with immunohistochemistry in our clinical TNBC cohort also confirmed a statistically significant OS advantage (logrank (Mantel-Cox) *p* = 0.0033; HR = 0.40 (0.21–0.75); Gehan–Breslow–Wilcoxon *p* = 0.0016; HR = 0.39 (0.20–0.73)) (Figure 2B). However, PFS data showed only a moderate tendency for a better outcome (logrank *p* = 0.1267; HR = 0.635 (0.35–1.14)). Furthermore, a significant OS benefit was observed in TNBC patients receiving chemotherapy compared to those who did not logrank *p* = 0.0072; HR = 0.36 (0.17–0.76) (Figure 2C). The AIF1 immunoreaction was cytoplasmic, often granular, suggesting mitochondrial localization in tumor cells with only very rare occurrence in cell nuclei, which would have indicated a direct induction of apoptosis. Typical examples of different AIF1 immunoscore categories are shown at low and high-power views in Figure 2D–G.

In support of these findings, the KM Plotter for breast proteins analysis revealed massive survival benefits for patients with high AIF1 protein (UniProt ID: 095831) expression in all aspects of tumor prognostic estimates (Figure 2H). AIF1 expression was linked to improved OS both at the best cutoff (*p* = 2 × 10^−5^; HR = 0.22 (0.1–0.47)) and at the median marker level separation (*p* = 0.00017; HR = 0.28 (0.13–0.57)) of the patients. The same was true both at the median and best cutoff separation values for RFS (*p* = 0.00096; HR = 0.38 (0.21–0.69)) and (*p* = 0.00012; HR = 0.27 (0.13–0.55)), and for DMFS (*p* = 0.00033; HR = 0.33 (0.17–0.62)) and (*p* = 3.7 × 10^−5^; HR = 0.3 (0.17–0.55)), respectively.

### 2.2. Expression of the Executioner Caspase, Caspase 3 in Relation with TNBC Prognosis

The most coherent impact of the tested biomarkers of the caspase-dependent apoptosis pathway on TNBC prognosis was noted for *CASP3* mRNA expression at the best cutoff values. RNAseq (NGS) data revealed a statistical correlation between high *CASP3* mRNA levels and OS (*p* = 0.042; HR = 0.44 (0.2–099)) (Figure 3A), while RNAchip data highlighted significant links of *CASP3* levels with better RFS (*p* = 0.045; HR = 0.67 (0.45–0.99)) (Figure 3B) and DMFS (*p* = 0.018; HR = 0.65 (0.45–0.93)) (Figure 3C), but not with OS. KM Plotter for breast protein analysis showed a strong positive trend between CASP3 (UniProt ID:P42574) levels and RFS but without significance (*p* = 0.065; HR = 0.38 (0.13–1.1)).

In line with the transcriptome results, in our clinical TNBC case series, immunohistochemistry revealed solid statistical correlations of cleaved/activated-CASP3 protein (c-CASP3) immunoscores both with better OS (logrank (Mantel-Cox) *p* = 0.0020; HR = 0.37 (0.20–0.70); Gehan–Breslow–Wilcoxon *p* = 0.0017; HR = 0.37 (0.20–0.68)) (Figure 3D) and PFS (logrank (Mantel–Cox) *p* = 0.0091; HR = 0.46 (0.26–0.83); Gehan–Breslow–Wilcoxon *p* = 0.0064; HR = 0.45 (0.45–0.80)) (Figure 3E). Furthermore, high in situ c-CASP3 expression could also be linked to significant OS advantage in chemotherapy-treated TNBC patients (logrank (Mantel–Cox) *p* = 0.011; HR = 0.38 (0.18–0.80); Gehan–Breslow–Wilcoxon *p* = 0.0071; HR = 0.37 (0.17–0.77)) (Figure 3F).

Cleaved/activated CASP3 immunoreactions occurred also in the tumor cell cytoplasm by covering similar but more moderate intensity ranges with a somewhat more frequent nuclear appearance, than of the AIF1 reactions. Typical examples representing diverse immunoscores are summarized in Figure 3G–I.

### 2.3. Expression of the Rest of Caspase-Dependent Apoptosis Biomarkers Tested and TNBC Prognosis

Of the other apoptosis-related biomarkers, only the mRNA expression of the anti-apoptotic *BCL2* showed obvious positive links with TNBC OS both at mRNAchip transcript (logrank *p* = 0.033; HR = 0.49 (0.25–0.96)) (Figure 4A) and protein levels (logrank *p* = 0.053 strong trend; HR = 0.37 (013–1.05)) (Figure 4B). These links were supported both for OS (logrank *p* = 0.008; HR = 0.39 (0.20–0.78)) (Figure 4C) and for PFS both in the whole cohort (logrank *p* = 0.014; HR = 0.45 (0.24–0.85)) and in the chemotherapy-treated patients (logrank *p* = 0.019; HR = 0.41 (0.20–0.86)) (Figure 4D) in our clinical TNBC cohort, when the strong expressor group (score 3) was compared to the low to medium expressors (scores1–2). However, a controversy was raised by an almost significant negative prognostic correlation of *BCL2* transcript expression with RFS (logrank *p* = 0.056; HR = 1.53 (0.99–2.39)).

The expression of *CASP8* mRNA (its protein is a potential mediator of extrinsic apoptosis signaling) was linked both with better RFS (*p* = 0.0031; HR = 0.57 (0.4–0.83)), DMFS (*p* = 0.0039; HR = 0.52 (0.33–0.82)) (Figure 5A,B) and OS (*p* = 0.052; HR = 0.52 (0.26–1.02)), though OS was only very close to significance. However, despite a moderate positive trend, cleaved/activated CASP8 protein expression (examples shown in Figure 5C) did not reveal any significant link with TNBC prognosis in our clinical cohort.

The prognostic role of the downstream trigger of caspase-dependent apoptosis cytochrome C, encoded by the *CYCS* gene, was the most controversial of all biomarkers tested. The KM Plotter RNAchip expression analysis of *CYCS* revealed statistically significant positive links with OS (*p* = 0.0068; HR = 0.70 (0.04–0.73)), but negative statistical correlation with RFS (*p* = 0.015; HR = 2.76 (1.17–6.49)) (Figure 6A,B). At the same time, apart from a moderate positive trend, in situ cytochrome c protein levels (examples shown in Figure 6C) did not show any significant link with TNBC prognosis in our clinical cohort.

### 2.4. Reproducibility of Immunoscoring and Correlations Between the Tested Biomarker Levels

The reproducibility of immunoscoring was tested using the interrater kappa-correlation, which showed strong significant agreements between the assessors (ranging between κ = 0.846 and 0.949); only the cCASP8 reaction was slightly below 0.8 (κ = 0.796) (Table 1). We also analyzed if correlations between biomarker levels may suggest functional links. Using both the Pearson and Spearman’s rank methods, significant positive correlations were found between cCASP3 and AIF1; CYCS and AIF1; and cCASP3 and BCL2 using both approaches (Table 2). However, kappa analysis revealed only their fair correlations: κ = 0.256, between AIF1 and CASP3; κ = 0.338, between AIF1 and CYCS; and κ = 0.205 between BCL2 and CASP3 (Table 3).

## 3. Discussion

TNBC is one of the most aggressive subtype of breast cancers with generally poor survival and widespread histopathological, transcriptomic and genomic diversity, which is only partly reflected in its molecular classification [6]. This fundamental heterogeneity may hinder the efficacy of the available treatment options [14]. In this study, we focused on the potential prognostic significance of the expression of major pathway elements of apoptosis [21], a type of programmed cell death response. Apoptosis can be induced, e.g., by treatment-related cell stress caused by radiotherapy [22] or chemotherapy (e.g., using doxorubicin, cisplatin or 5-fluorouracil), which trigger DNA damage and mitochondrial dysfunction [23]. Our results show that testing for the expression of BCL2 and the executioner apoptotic factors AIF1 (gene *AIFM1*) and caspase 3 (gene *CASP3*) may identify partly overlapping TNBC subgroups with favorable prognosis. These findings may offer rationale for future studies evaluating the potential of apoptosis-targeting treatment strategies in this aggressive tumor.

In TNBC treatment, several regimens exploit the activation of apoptotic pathways for tumor cell elimination. Besides radiation therapy and conventional chemotherapy [22,23] several targeted approaches have been utilized for apoptosis induction. TRAIL (TNF-related apoptosis-inducing ligand) and its receptor agonists showed promising results in preclinical studies of TNBC, particularly in cell lines with mesenchymal features [24,25]. BH3 mimetics, such as BCL2-xL inhibitors, have demonstrated synergistic effects when combined with standard chemotherapy in TNBC models [26]. The BCL2 inhibitor venetoclax may also be used for apoptosis induction in TNBC of elevated target expression [27]. Novel compounds such as magnolol were found to induce both extrinsic and intrinsic apoptotic pathways in TNBC cells while simultaneously suppressing EGFR/JAK/STAT3 signaling [28]. The development of combination therapies, such as birinapant (a SMAC mimetic) with gemcitabine, also showed enhanced antitumor efficacy by overcoming apoptosis resistance in TNBC [29].

The apoptosis-inducing factor-1, encoded by the AIFM1 gene, is an oxidoreductase flavoprotein of the mitochondrial respiratory complexes. When released from mitochondria, it can trigger endonuclease activation and controlled DNA cleavage, acting as a mediator of caspase-independent apoptosis [30,31,32,33]. Our in silico analysis revealed that elevated AIFM1 mRNA expression can be linked to significantly improved OS, but did not reach significance either with RFS or DMFS. This suggested a selective protective effect on long-term survival rather than tumor recurrence or metastasis. This observation was supported at the in situ protein level in our clinical TNBC cohort of relatively limited sample size (103 patients). Immunohistochemistry also confirmed that tumors with higher AIF1 protein expression were associated with significantly better OS, but showed only a moderate tendency for better PFS. Further backing of these results came from KM Plotter for breast proteins analysis. This reverse-phase protein array (RPPA)-based test revealed massive survival benefits for patients of elevated AIFM1 protein expression not only for OS but also for RFS and DMSF. These associations remained significant even when using strict median-based thresholds to separate high and low expression groups. The controversial link of AIFM1 expression with RFS and DMSF at the protein and mRNA level may be explained by the distinct cohorts used in the different analyses. Nevertheless, all these tests reinforced that AIF1 expression can serve as a favorable prognostic marker in a subpopulation of TNBC.

In line with our findings, a proteomic study, using nanoscale liquid chromatography and tandem mass spectrometry also identified AIFM1 as part of an 11-protein signature associated with favorable DMFS in TNBC [34]. Upregulated AIFM1 was linked with a reduced hazard of TNBC relapse or distant metastasis within 5 years. Furthermore, high AIFM1 expression was related to better outcome, while reduced expression resulted in poorer patient OS across multiple cancer types [35].

Since chemo- and radiotherapy often induce a programmed cell death response, we also tested how AIF1 expression affected TNBC patients’ response to therapy. Indeed, we found that patients with AIF1-high tumors responded significantly better to standard chemo-/radiotherapy or chemotherapy than those with AIF1-low cancers. Although in TNBC cell lines the inhibition of the *AIFM1* gene was shown to increase chemosensitivity [36], we are convinced that the results gained in clinical human tumors surrounded by their complex microenvironment better represent clinical situations. The subcellular distribution of AIF1 protein reflects its dual functionality [30]. As a flavoprotein, AIF1 normally localizes to the mitochondrial intermembrane space as part of the oxidative phosphorylation and cellular redox balance chain [30]. However, upon cell stress, it is released from the mitochondria and then translocates to the nucleus to trigger apoptosis [30]. In our primary TNBC cohort, we mainly observed AIF1 protein in mitochondrial localization without major nuclear translocation in line with their therapy naïve status (only nine cases received neoadjuvant chemotherapy). On the other hand, most malignant tumors shift their energy generation metabolism to oxidative glycolysis [37,38], which may be hampered by the elevated AIF1 level in mitochondria. Therefore, it seems that the availability of AIF1 protein mainly in the mitochondria may both support caspase-independent apoptosis upon chemotherapy and reduce the tumor promoting dominance of oxidative phosphorylation. These parallel antitumor effects may explain the prognostic benefits of AIF1 expression in TNBC.

Of the caspase-dependent apoptotic cascade, also the executioner member, CASP3 showed positive associations with TNBC prognosis. High *CASP3* mRNA expression correlated with significantly improved OS in RNAseq data, and with better RFS and DMFS in RNAchip datasets in KM Plotter testing. RPPA-based KM Plotter for breast protein analysis indicated only a strong positive trend with RFS, without major link with OS or DMFS. However, our in situ-detected cleaved/activated CASP3 protein levels using immunohistochemistry revealed robust positive statistical correlations both with prolonged OS and PFS. Elevated cCASP3 levels in situ also demonstrated major survival benefits for radio/chemotherapy-treated TNBC patients. These findings emphasize that the prognostic value of cCASP3 within TNBC is the strongest when evaluating its activated/cleaved form of the protein. This underscores the potential utility of immunohistochemistry in assessing apoptosis-related biomarkers, though further research and validation are required to determine its clinical value.

In support of our findings, recent studies also highlighted the tumor-suppressive effect of activated CASP3 in TNBC. Elevated cCASP3 levels induced by gallic acid treatment promoted apoptosis and suppressed TNBC cell proliferation [39]. Likewise, tetra-arsenic hexoxide could also trigger CASP3 dependent cell death in HCC1806 TNBC cells in vitro, through cleaving gasdermin E [40]. However, our results challenge some other reports. In advanced TNBC cases, despite cCASP3 expression predicting improved chemotherapy response, paradoxically, it also indicated worse overall survival [41]. In addition, in a meta-analysis testing all subtypes of breast cancer, CASP3 overexpression was linked to shorter survival [42]. The reasons for these controversies potentially lie in the distinct TNBC cohorts, chemo- and/or radiotherapy regimens or specificity of CASP3 (cleaved or non-cleaved) antibodies used. These may also highlight the context dependent prognostic role of CASP3. The proportions of neoadjuvant treated cases, which were only sporadic (nine patients) in our clinical cohort, as well as the lack of focusing only on the TNBC subtype, should also be considered. Nevertheless, our results suggest that in TNBC, both AIF1 and the executioner cCASP3 in stock may contribute to the programmed cell death response following adjuvant chemo-/radiotherapy to support better patient outcome. However, the confirmation of this needs further functional studies.

Caspase-dependent apoptosis can be induced both through the extrinsic, death receptor (FAS, DR4 or DR5)-mediated pathway and the intrinsic route via mitochondrial stress [43]. CASP8 expression, which can promote both pathways, showed a significant survival advantage at the transcript level both with RFS and DMFS, besides a positive trend with OS. However, in situ cleaved/activated CASP8 protein expression did not demonstrate any major prognostic impact in our clinical TNBC cohort, apart from a moderate positive trend. Factors such as epigenetics-related post-transcriptional regulation and protein turnover may explain the discrepancy between the effects of mRNA and the cleaved-activated protein expression. Our findings may also be explained by the context-dependent dual role of CASP8. In cancer, silencing of the CASP8 gene in an MDA-MB-231 TNBC cell line reduced cell growth but caused a more aggressive and motile phenotype [44]. Accordingly, in our clinical cases, high CASP8 levels might not restrict tumor cell proliferation, but potentially reduce TNBC cell migration and metastatic spread, resulting in improved RFS and DMFS. Nevertheless, our data suggest that the extrinsic pathway may not play a leading role in executioner CASP3 recruitment in cancer cell eliminations in TNBC.

Remarkably, we found the expression of the mitochondrial anti-apoptotic BCL2, which controls the pro-apoptotic BAX and BAK [45,46], to also show significant positive links with TNBC prognosis. Elevated *BCL2* mRNA expression was connected to significantly improved OS. Likewise, the RPPA-based KM Plotter for breast protein assay revealed a strong, nearly significant trend with OS. Furthermore, high in situ BCL2 protein expression was massively associated both with prolonged OS and PFS in our clinical cohort, where OS benefit was also evident in chemotherapy-treated patients. In line with our findings, in a comprehensive meta-analysis of over 5000 cases, BCL2 immunohistochemistry concluded BCL2 as an independent positive prognostic factor across breast cancers [47]. High BCL2 protein expression was also linked by others to improved disease-free survival in breast cancer [48]. On this line, BCL2 expression was also often associated with features of less aggressive tumor biology [49].

However, this favorable prognostic profile in our TNBC study has been obscured by a nearly-significant negative trend between *BCL2* transcript levels and RFS. Since these data were gained using the same cohort and testing platform, our results suggested a possible early relapse tendency of *BCL2*-high tumors. However, some conflicting reports on the prognostic impact of BCL2 in TNBC were also published. Though in a large TNBC cohort BCL2-positive cases also showed markedly better prognosis than BCL2-negative ones, the latter group responded better to anthracycline-based chemotherapy [50]. In another work, BCL2 expression was a predictor of worse outcome in TNBC [51], while in a subtype-specific breast cancer study, BCL2 status did not significantly influence the 5-year survival in TNBC. At the same time, BCL2 expression proved to be prognostic in the luminal A subtype stating that BCL2 is an estrogen-responsive gene [49]. All these discrepant observations including ours highlight the importance of the cohort- and therapy-related context dependent molecular heterogeneity of TNBC. Taken together, our results indicate that BCL2 expression may serve as a favorable prognostic factor in TNBC, at least for long-term survival outcomes.

Of our tested biomarkers, CYCS, a mitochondrial oxidoreductase, which may contribute to CASP3 activation [52], showed the most controversial prognostic profile in TNBC. At the transcript-level *CYCS* expression showed a significant positive correlation with OS, but also a massive negative link with RFS. Meanwhile, in our clinical tumors CYCS protein expression revealed statistically meaningless trends. These controversies are also reflected by others’ findings. On one hand, it was demonstrated that activation of the CYCS–CASP9 apoptotic pathway by berberine effectively suppressed TNBC cell growth both in vitro and in vivo [53]. On the other hand, elevated *CYCS* transcript levels were associated with reduced OS and increased metastasis risk in a mixed breast cancer cohort of all subtypes [53]. Additionally, in a TNBC subset of breast cancers of all subtypes, low CYCS and high VDAC1 (voltage-dependent anion channel-1) protein levels were linked significantly to reduced 5-year DFS [54]. These divergent results highlight the complex biological role of CYCS and the context, cohort (molecular subtype, grade, stage) and test dependence of its study results in breast cancer. Furthermore, the steady-state CYCS level in breast cancers is excessively reduced compared to healthy tissue, which may hinder the accurate assessment of its expression in cancer [55]. For a pathobiochemical explanation of this prognostic controversy, a study suggested that only the oxidized CYCS can effectively promote apoptosis, while its reduced form is inactive, emphasizing the importance of its redox state besides the expression level [56].

The positive statistical rank correlations between AIF1 and CASP3, AIF1 and CYCS and between BCL2 and CASP3 in situ protein expression and the significant agreement in assessors’ scores suggested their functional links and the potential of immunohistochemistry in their prognostic testing. However, the only fair kappa correlations found in individual cases (kappa = 0.205–0.338) revealed their modest direct links in individual samples. On the other hand, there were some signal alternatives but partly interacting pathways, which also affect their functions [18]. For example, the highest, almost good correlation was found between AIF1 and CYCS (correlates in 69 vs. 34 patients), two cell respiratory oxidoreductases. They are released after mitochondrial outer membrane damage upon cell stress, and activate mainly alternative signaling. AIFM1 and cCASP3 (correlates in 64 vs. 39 patients) may function at the executioner end-points of apoptosis through partly alternative pathways [33]. BCL2 inhibits mitochondrial membrane damage induced by pro-apoptotic factors that may activate CASP3 (correlates in 65 vs 38 patients). Despite opposing functions, even their modest positive prognostic association reflects their complex regulation. However, the molecular heterogeneity of TNBC subtypes calls our attention to the context and cohort dependence of all prognostic biomarker studies [6].

In this study, we observed differing prognostic strength of some biomarkers in the same TNBC cohort at different disease end-points (OS, RFS or DMFS). This did not raise major concerns if the same platform was used and could be explained, e.g., by differences in treatment dynamics and duration up to these end-points. However, occasionally mRNA and protein abundance showed non-linear or even discordant prognostic correlations, experienced also by others in complex samples [57,58]. In general, technological, post-transcriptional and post-translational modifications can contribute to the discrepant mRNA and protein levels observed; e.g., in our testing of the prognostic role of CASP8 and CYCS expression [59,60]. The translation efficiency of mRNA abundance mainly depends on its stability/speed of decay besides microRNA regulation, which latter can silence mRNA translation [61]. On the other hand, post-translational mechanisms can regulate protein function and detectability, including, e.g., cleavage, phosphorylation, ubiquitination, acetylation or methylation [60]. Furthermore, method-related technical factors such as antibody sensitivity and fixation-related antigen preservation can also influence the efficiency of in situ protein detection with immunohistochemistry [62]. All these may contribute to the occasionally inconsistent results of biomarker studies. Further standardization of pre-analytics and detection methods, as well as molecular studies on target gene transcription and translation, are still needed to clarify the role of these modifying factors.

It is important to note that most of our clinical TNBC patients were therapy naïve at surgery/sample collection. Only nine received diverse neoadjuvant therapies, which, however, resulted in no noticeable pathological tumor regression by imaging follow up. Therefore, they were also included in this prognostic study and received similar therapy to the rest of the 94 TNBC cases. The chemotherapy regimens administered to our TNBC cohort were heterogeneous, including various combinations of anthracyclines, taxanes, platinum agents and other drugs (anthracycline-based: 73 patients, taxane-containing: 65 patients, platinum-based: 3 patients, 5FU-containing: 16 patients, avastin-containing: 1 patient), with or without radiotherapy (Table 4). While this reflects real-world clinical practice, it also introduces treatment-related variability that can influence survival outcomes. Unfortunately, our sample size relative to the diversity of treatment protocols did not allow stratified analyses by specific treatment type. This needs to be tested in the future in large multicenter TNBC cohorts.

In conclusion, the insufficient programmed cell death response is one of the major hallmarks of cancers [63], which generally affects breast cancers including the aggressive TNBC subtype. In this study, focusing on major elements of the apoptosis pathways, we demonstrated that in primary, untreated TNBC, the elevated expression of the executioner apoptotic factors AIF1 and CASP3, and of BCL2 can be linked to significant survival advantage, both at the mRNA and protein level. The biomarker levels only moderately overlapped in individual patient samples, which underline the complex regulation of apoptotic pathways. In the primary tumors, AIF1 and cleaved CASP3 proteins only rarely showed nuclear translocation. This suggested that therapy may recruit them from the cytoplasmic/mitochondrial stocks in proportion to their expression to contribute to improved patient survival in TNBC. Accordingly, testing for AIF1, cleaved CASP3 and BCL2 expression, e.g., using immunohistochemistry, may identify partly overlapping TNBC subgroups with improved survival. However, the clinical utility of apoptosis-targeting strategies remains to be established in further studies. Despite our promising results, we acknowledge several limitations of our study, including its single-center and retrospective design, the absence of an independent external validation cohort, the lack of functional validation and the moderate sample size. These may limit the generalizability and robustness of our findings. Therefore, future studies involving larger, multi-institutional cohorts and mechanistic investigations are required to endorse our observations.

## 4. Materials and Methods

### 4.1. In Silico mRNA and Protein Expression Analysis in Triple-Negative Breast Cancer

First, we analyzed the prognostic value of apoptosis-related biomarker expression in TNBC in silico both at the mRNA and protein levels using an open-source database management software package, Kaplan–Meier Plotter (http://kmplot.com) [64]. Transcript expression data were obtained from the Gene Expression Omnibus, GEO [65] and EGA [66] databases including mRNAchip and next-generation sequencing (NGS)-based RNAseq data from 4929 and 2976 breast cancer patients, respectively, including TNBC. Protein expression data were also available in this database, which were gained by reverse-phase protein array (RPPA) of The Cancer Proteome Atlas (TCPA v3.0; https://tcpaportal.org/tcpa/, accessed on 14 May 2025), a functional proteomics database with data from 221 TNBC patients [67].

Our survival analyses focused on overall survival (OS), relapse-free survival (RFS) and distant metastasis-free survival (DMFS). The breast cancer cohorts were filtered for ER-negative, PR-negative and HER2-negative cases, resulting in data from 392 TNBC patients for transcriptome analysis. Patients were stratified into high and low expressors by dichotomizing each cohort either at median biomarker expression values, or at the auto-selected best cut-off thresholds offered by the Kaplan–Meier Plotter algorithm. We also tested the prognostic impact of the treatment in relation to the expression of our biomarkers by comparing the survival data of systematically untreated and treated patients.

### 4.2. Triple-Negative Breast Cancer Cohort Used for Testing In Situ Protein Expression

TNBC cases, characterized by the absence of ER, PR and HER2 expression, were diagnosed and collected at the Pathology Department of the Bács-Kiskun County Teaching Hospital, Kecskemét, Hungary. Formalin-fixed and paraffin-embedded tissue blocks from 110 consecutive, primary TNBC cases were selected for this study, which was performed in line with the regulations of the WMA Declaration of Helsinki and was approved by the National Committee for Research Ethics (ETT TUKEB) under the permission No. BMEÜ/443-5/2022/EKU issued in 10/10/205. The Committee waived the need for individual patient consent on archived tissues available for primary diagnostics for the purpose of retrospective biomarker testing.

Tissue microarrays (TMAs) were constructed from the selected TNBC donor paraffin blocks using a computer-driven microarray builder TMA Master (3DHistech Ltd., Budapest, Hungary). Duplicate cores of 2 mm diameter were collected from representative areas of the tumors, which were assembled into three pieces of 70-sample TMA blocks, totaling 210 samples. Some TMA cores also contained normal/reactive breast glands (terminal duct lobular units), and a few cores of reactive tonsil included at random positions in the TMA blocks served for technical controls of antibody specificity and evenness of staining.

After leaving out some cores, which contained either damaged or too small tumor tissue, our study cohort finally consisted of tumor samples from 103 patients with a median age of 69 years (range: 29–91 years) at diagnosis. The major clinico-pathological data of the TNBC cases tested in this study are summarized in Table 4. It is important to note, that most TNBC samples were therapy naïve, only nine cases received neoadjuvant chemotherapy, but without recognizable regression by imaging follow up. Therefore, all 103 patients received adjuvant therapy and their prognostic data were considered accordingly.

### 4.3. Immunohistochemistry

Serial sections of 4 µm thickness were cut from TNBC TMAs for immunohistochemical staining in the Roche-Ventana Benchmark automated system (Roche-Ventana, Tucson, AR, USA). Briefly, after dewaxing the slides by the instrument, antigen retrieval was carried out using the pH 9.0 (CC1) buffer for 60 min, followed by incubation with monoclonal rabbit anti-cleaved-CASP3 (#9664, Asp175, clone5A1E; 1:200); anti-cleaved-CASP8 (#8592, Asp387, clone D5B2; 1:200); anti-CYCS (#4280, clone 136F3; 1:100); monoclonal mouse anti-BCL2 (clone 124; 1:50); and polyclonal mouse anti-AIF1 (#4642; 1:50) primary antibodies for 60 min each. Antibody binding was detected using the Ultraview polymer peroxidase kit (Roche-Ventana) and the immunoreactions were visualized with the diaminobenzidine tetrahydrochloride (DAB, brown) hydrogen peroxide chromogen-substrate kit (Roche-Ventana). All antibodies were from Cell Signaling (Danvers, MA, USA), except the anti-BCL2, which was obtained from Agilent-Dako (Glostrup, Denmark).

### 4.4. Scoring of Immunoreactions

Hematoxylin and Eosin (H&E) stained and immunostained slides were digitalized using a Pannoramic 1000 Flash DX scanner and scored in the SlideViewer program (both 3DHistech). All immunoreactions except for BCL2 were scored using an Allred-type scheme, which combined the percentage of positive cells (score 1 = <5%; 2 = 6–20%; 3 = 21–41%; 4 = 41–60%; and 5 = >60%) with the intensity (score 1 = low, 2 = medium and 3 = strong) of the reaction product in the tumors, resulting in a score range between 2 and 8. Only few immunoreactions were considered negative for BCL2, but none for the other tested biomarkers. Since BCL2 immunostaining showed rather homogenous intensities and distribution, it was scored on a simplified scale, where score 0 = no obvious reaction/negative, score 1 = low-, score 2 = medium- and score 3 = high expression. After setting up and agreeing on the evaluation criteria, the immunoreactions were independently scored by a pathologist (AN) and a senior scientist (TK) blinded to the clinicopathological data of TNBC patients. For survival estimations, the final scores were consolidated for dichotomization by involving a second pathologist assessor (MT).

### 4.5. Statistical Analysis

In silico transcript and protein expression-related prognostic (OS, RFS and DMFS) correlations were gained using the built-in functions of the Kaplan–Meier Plotter [68]. The survival estimates of our immunohistochemistry-based biomarker expression data were tested using the GraphPad Prism v8.0.1 software (Boston, MA, USA). Patient follow-up and treatment-related data were collected and extracted for analysis from our digital patient registration xls. file. Patients were alive, lost to follow-up or were censored at the time of their last visit. Survival curves were generated both for the TNBC patients related in silico and in situ biomarker expression data and the log-rank test was applied to assess statistical significance between high and low expressor groups. Hazard ratios (HR) with 95% confidence intervals (CI) were calculated to quantify the association between biomarker expression and survival outcomes. For calculating the interrater agreement kappa values as well as the Pearson and Spearman’s rank correlations between the pairs of apoptosis related biomarker levels, IBM SPSS V.16 Statistics (Armonk, NY, USA). A *p*-value < 0.05 was considered statistically significant for all analyses.

## Figures and Tables

**Figure 1 ijms-26-07227-f001:**
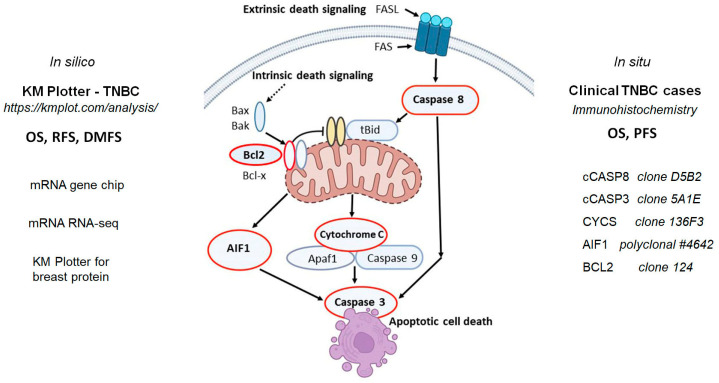
Study design showing a simplified model of apoptotic pathways and the related biomarkers tested (encircled in red). First, we analyzed the prognostic impact (OS, RFS and DMFS) of the apoptosis-related biomarker gene and protein expression in silico in TNBC cases, focusing on apoptosis-inducing factor-1 (AIF1, gene AIFM1), caspase 8 (CASP8), caspase 3 (CASP3), Cytochrome C (CYCS) and BCL2 (BCL2) using the KM Plotter database (https://kmplot.com/analysis/, accessed on 2 May 2025). Then, the survival data of our clinical TNBC cohort were correlated with the in situ protein expression results gained using immunostaining for the same biomarkers involving the cleaved/activated forms of caspases, cCASP3 and cCASP8.

**Figure 2 ijms-26-07227-f002:**
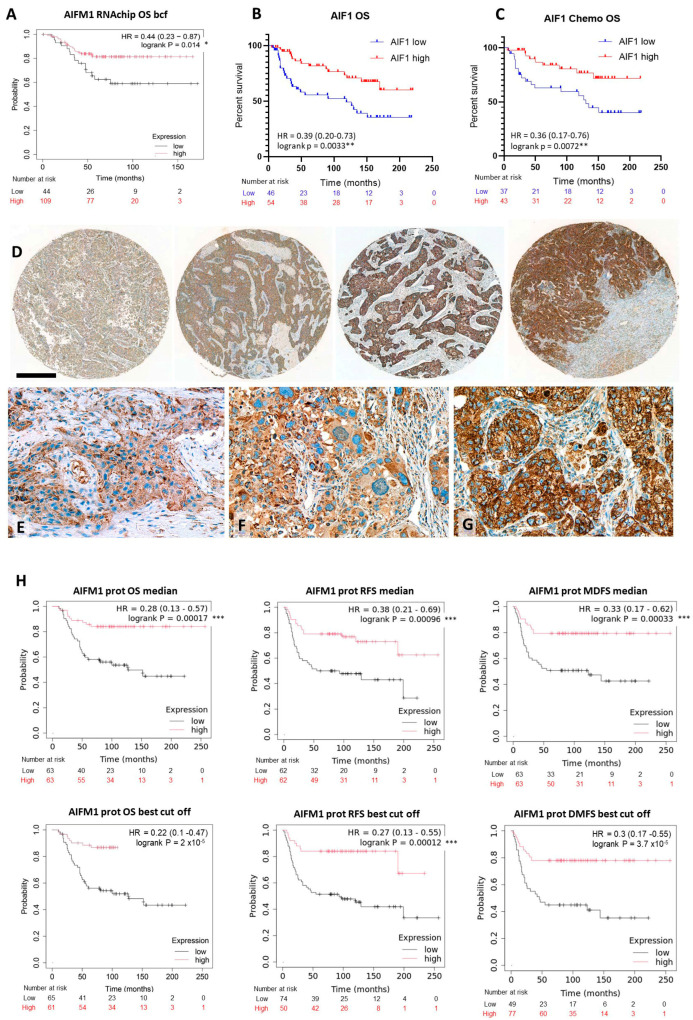
Elevated *AIFM1* mRNA expression showing significantly better OS in TNBC cases with KM Plotter analysis (**A**) confirmed by the immunostaining results in our clinical cohort (**B**). The latter was also linked to OS benefit for chemotherapy treated vs. untreated patients (**C**). A series of TNBC TMA cores demonstrating diverse AIF1 protein expression in situ (**D**). High power images show examples of score 3 (**E**), score 4 (**F**) and score 8 (**G**) of AIF1 protein expression with weak diffuse (**E**,**F**) vs. strong granular cytoplasmic reactions (**G**). DAB immunoperoxidase (brown). Significance: *p* < 0.05 *; *p* < 0.01 **; *p* < 0.001 ***. Scale bars on (**D**): 200 µm represents 50 µm on (**E**–**G**). KM Plotter for breast proteins analysis also confirms highly significant survival benefits of high AIF1 protein levels at all endpoints (OS, RFS and DMFS) in TNBC patients (**H**).

**Figure 3 ijms-26-07227-f003:**
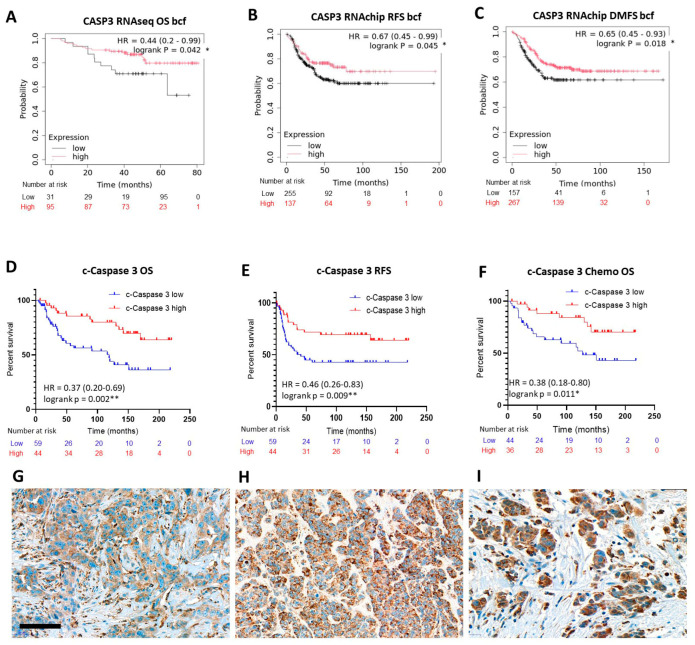
Elevated *CASP3* mRNA expression significantly linked with improved prognosis, OS (**A**), RFS (**B**) and DMFS (**C**) in TNBC cases with the KM Plotter analysis. Better patient outcomes both for OS (**D**) and PFS (**E**) were supported by high immunoscores of in situ CASP3 protein expression in our clinical cohort. The latter is also associated with a significant OS benefit for chemotherapy-treated TNBC patients (**F**). Significance: *p* < 0.05 *; *p* < 0.01 **. A series of TNBC samples demonstrating score 3 (**G**), score 6 (**H**) and score 7 (**I**) CASP3 immunostaining results. DAB immunoperoxidase (brown). Scale bar, (**G**–**I**): 50 µm.

**Figure 4 ijms-26-07227-f004:**
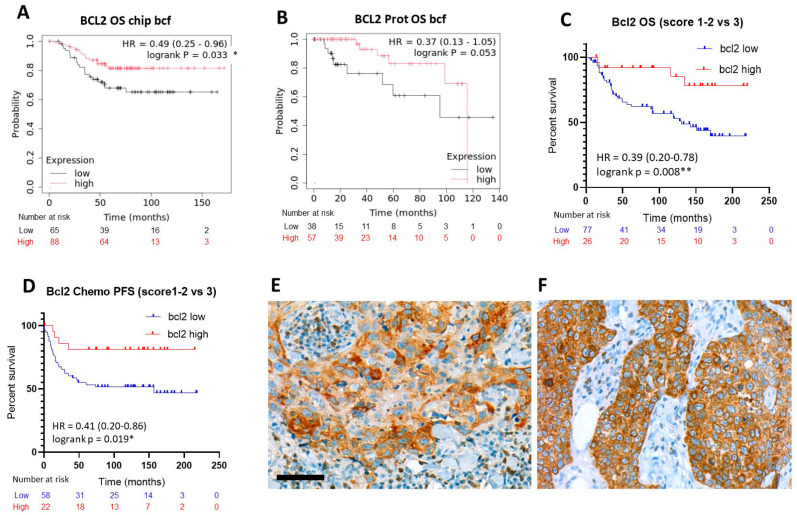
Elevated BCL2 transcript and protein expression showing significantly better OS in TNBC with the mRNAchip (**A**) and the breast protein ((**B**), close to significance) analysis, respectively, using the KM Plotter algorithm. The survival advantage confirmed also in our clinical cohort both at OS (**C**); and at PFS (**D**), both in the whole cohort and in the chemotherapy-treated patient group (**D**). Examples of score 2 (**E**) and score 3 (**F**) cytoplasmic BCL2 protein expression in TNBC tissue sections. DAB immunoperoxidase reactions (brown). Significance: *p* < 0.05 *; *p* < 0.01 **. Scale bar, (**E**,**F**): 50 µm.

**Figure 5 ijms-26-07227-f005:**
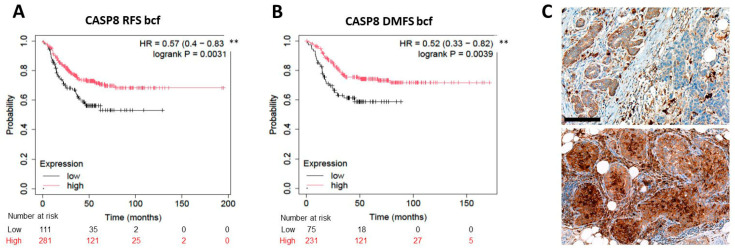
Elevated *CASP8* mRNA expression significantly linked with improved RFS (**A**) and DMFS (**B**) in TNBC cases in KM Plotter analysis. Significance: *p* < 0.01 **. Low ((**C**), upper panel) and high cytoplasmic cleaved/activated CASP8 expression in the tumor nests ((**C**), lower panel) revealed using immunohistochemistry. DAB immunoperoxidase (brown). Scale bar, (**C**): 50 µm.

**Figure 6 ijms-26-07227-f006:**
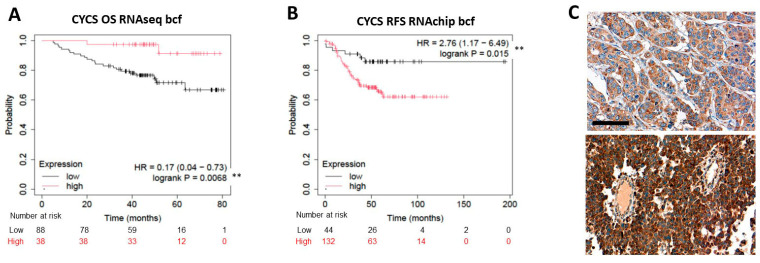
Controversial, significantly opposing correlations of high and low CYCS mRNA expression with OS (**A**) and RFS (**B**), respectively, in TNBC analyzed using the KM Plotter. Significance: *p* < 0.01 **. Medium ((**C**), upper panel) and high cytoplasmic expression ((**C**), lower panel) of CYCS immunoreactions. DAB immunoperoxidase (brown). Scale bar, (**C**): 50 µm.

**Table 1 ijms-26-07227-t001:** Interrater kappa-correlations between assessors scores of immunoreactions.

Marker	Kappa (κ)	*p*-Value
AIF1	0.949	<0.001
BCL2-2	0.856	<0.001
cCASP3	0.846	<0.001
cCASP8	0.796	<0.001
CYCS	0.852	<0.001

**Table 2 ijms-26-07227-t002:** Correlations between the expression of apoptosis-related biomarkers detected with immunohistochemistry using both the Pearson and Spearman’s rank methods.

Marker Pair	Pearson Correlation	Pearson *p*-Value	Spearman’s Correlation	Spearman’s *p*-Value
cCASP3-AIF1	0.260	0.008 *	0.277	0.005 *
cCASP8-AIF1	0.143	0.150	0.126	0.206
BCL2-AIF1	0.088	0.375	0.0940	0.345
CYCS-AIF1	0.438	<0.001 *	0.459	<0.001 *
cCASP3-cCASP8	0.126	0.203	0.089	0.371
cCASP3-BCL2	0.250	0.011 *	0.255	0.009 *
cCASP3-CYCS	0.077	0.438	0.039	0.697
cCASP8-BCL2	−0.002	0.980	0.029	0.768
cCASP8-CYCS	−0.019	0.847	0.021	0.827
BCL2-CYCS	0.021	0.831	0.017	0.863

* Statistically significant positive link

**Table 3 ijms-26-07227-t003:** Kappa correlations between apoptosis-related biomarker pairs found significant in Table 2.

			Correlations	Cases
	cCASP3-low	cCASP3-high		
AIF-low	33	13	Kappa = 0.254 SE = 0.092 95% CI: 0.074–0.434	46
AIF-high	26	31		57
Total	59	44	64 vs. 39 correlate	103
	CYCS3-low	CYCS-high		
AIF-low	31	15	Kappa = 0.338 SE = 0.093 95% CI: 0.156–0.519	46
AIF-high	19	38		57
Total	50	53	67 vs. 36 correlate	103
	cCASP3-low	cCASP3-high		
BCL2-low	49	10	Kappa = 0.205 SE = 0.092 95% CI: 0.025–0.385	59
BCL2-high	28	16		44
Total	77	26	65 vs. 38 correlate	103

**Table 4 ijms-26-07227-t004:** Summary of clinico-pathological data of TNBC patients tested.

Number of patients	103
Histology	invasive ductal carcinoma
Median age (at case diagnosis)	69 years (range: 29–91 years)
Median follow-up	98 months (range: 5–220 months)
Endpoint category	LFU-AWD: 13, LFU-NED: 13, AWD: 2, AWOD: 2 DOD: 13, DOOC: 14, DWD: 3, NED: 28
Median mortality	65 months (range: 5–170 months)
Median progression	26 months (range: 1–157 months)
Tumor size	22.1 mm (range: 7–55 mm)
Neoadjuvant chemotherapy	9
pT Category	pT1: 51, pT2: 43, pT3: 4, pT4: 5
pN Category	pN0: 54, pN1: 35, pN2: 10, pN3: 2, pNx: 1
Grade	G1: 0, G2: 4, G3: 99
Surgery type	Breast conservative: 77, Mastectomy: 26, SNB: 52, SNB + ABD: 23, ABD: 26, No axillary surgery: 2
Chemo and Radiotherapy	Only chemotherapy: 9, Chemo- and radiotherapy: 72, Only radiotherapy: 9, Neither chemo- nor radiotherapy: 13

Abbreviations: LFU-AWD: Lost to Follow-Up–Alive with Disease, LFU-NED: Lost to Follow-Up–No Evidence of Disease, AWD: Alive with Disease, AWOD: Alive Without Disease, DOD: Died of Disease, DOOC: Died of Other Causes, DWD: Died with Disease, NED: No Evidence of Disease, SNB: Sentinel Node Biopsy, ABD: Axillary Block Dissection.

## Data Availability

Data supporting reported results are available on reasonable request from the corresponding author.

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
