# Peer review of "Prognostic Potential of Apoptosis-Related Biomarker Expression in Triple-Negative Breast Cancers"

_ijms, 2025, doi:10.3390/ijms26157227_

Round 1
Reviewer 1 Report
Comments and Suggestions for Authors
This manuscript addresses the prognostic value of apoptosis-related markers in triple-negative breast cancer (TNBC) using both in silico transcriptomic data and immunohistochemical (IHC) evaluation in a clinical cohort. The subject is both timely and clinically relevant. However, several significant concerns must be addressed before this work is suitable for publication.
1 The conclusions occasionally overstate the clinical impact of the markers evaluated. For instance, suggestions that these markers may guide “tailored treatment strategies” are not supported by any therapeutic or mechanistic data. These assertions should be reframed to reflect the exploratory and correlative nature of the findings.
All clinical findings are based on a single retrospective cohort without external validation. Additional independent datasets or cross-validation strategies would be needed to confirm generalizability. Moreover, the rationale for focusing exclusively on this set of markers should be more clearly articulated in the context of existing literature.
3 Several markers show discordance between mRNA expression and protein localization or abundance. These inconsistencies should be more thoroughly addressed in the Discussion to clarify their implications for biomarker development.
4 The manuscript would benefit from more detail regarding:
-The patients’ treatment backgrounds (e.g., chemotherapy status, endocrine therapy)
-The timing of tissue collection relative to treatment
-The exact cutoffs used in IHC scoring
-Whether any batch effects or interobserver biases were accounted for during scoring
5 The figures are described as informative but could benefit from improved clarity, including more descriptive legends and consistent labeling. A heatmap or summary visualization comparing mRNA and protein expression trends could add clarity.
6 The Discussion would benefit from a more candid reflection on the study’s limitations, including the retrospective design, lack of functional validation, and moderate sample size.
The English is generally understandable but includes numerous awkward phrases and overly long sentences. A professional English language edit is recommended to improve fluency, especially in the Results and Discussion sections.
Author Response
Thank you for your recommendations any criticisms!
We realized that in the uploaded manuscript some needless hyphens remained through the copying process. Also, some mistyping, typographical errors and fragments were found from the earlier version in our manuscript. All these are corrected a few expressions are replaced, and some sentences are shortened or were broken into two to support easier comprehension of our results and their interpretation. We accept that our data include multifaceted analyzes, which may not be easy to follow, but need to be presented.
Also, in the 2nd sentence of our introduction, we refined the breast cancer subtype nomenclature according to the most recent version.
- We deleted the word “tailored” from our conclusion to make it more modest: “…may identify a patient subgroup within TNBC, which potentially benefit from apoptosis-targeting treatment strategies.” Also deleted the last concluding sentence from the introduction.
We agree that the apoptosis pathway is more complex, but due to our immunohistochemistry approach, which is available for most pathology labs, we focused on some of its major crossroad markers including executioner elements. Fig. 1, the 4th paragraph of our introduction and (after relocation) the 2nd paragraph of our discussion we gave a focused explanation both for the links among these markers and their relevance in apoptosis testing and therapy. Prospective collection of further patient samples from our diagnostic files is under way for validating our results and searching for other relevant markers in the future.
- We also agree that mRNA and protein levels can be controversial due to several reasons. This was partly highlighted in the discussion of BCL2 results, which we completed in the revised manuscript (see lines 400-404). Nevertheless, our significant markers AIFM1, CASP3 showed good correlations between mRNA and protein expression.
There is also an explanation on this including functional issues in the discussion at CYCS results (see lines 418-425).
- Clinicopathological data of our TNBC patients including treatment details are summarized in our Table 3. in the Materials and Methods. They are all estrogen receptor negative, therefore, they did not get hormone treatment. The vast majority of our patient samples are treatment naive at diagnosis/sample collection, only 9/103 received neoadjuvant chemotherapy, but survival data do reflect the effect of therapies. This is mentioned in the text but we also emphasize it under the 4.2 subtitle in the Mat&Meth and additionally included this into our Table 3. Cutoffs in IHC scoring are detailed under the 4.4. subtitle in the same chapter. Interrater kappa values are also detailed in Table 1. of the Results (see also lines 491-493).
- Obviously presenting the relevant data in the complex, multifaceted analyzes throughout the manuscript and particularly in the Results section, may not easy to follow. We wanted to provide detailed but focused figure legends, which together with the body text reflect all relevant details, but avoid full repetition of body text.
- We included a sentence as you suggested on our study limitations: “Despite our promising results, we have to mention the limitations of our study, which include the retrospective design, lack of functional validation, and relatively moderate sample sizes.”
We used the professional language of cancer diagnostics and research. A few words in the original version are replaced in the revised version, but if you find any awkward wording please specify to help us improve.
Reviewer 2 Report
Comments and Suggestions for Authors
Comments to authors:
The manuscript of a research article, which was written by Miklós Török et al, is interesting, discussing correlations between expressions of apoptosis-inducing factors and overall survival of TNBC patients. The conclusion would be important if the obtained results could be applied to the estimation of chemotherapeutic drugs. However, I suggest authors edit text and discuss more about the results obtained before publication.
Recommendation: Minor revision
General comments
As shown in Table 3, most of the data were obtained from patients taking Chemo- and Radiotherapy. Readers will wonder if high levels of AIF1/CASP3/CYCS/BCL2 are indicated in biopsy of Chemo- and Radiotherapy taking patients. Authors might as well confirm if some specific anticancer drugs can raise the apoptotic factor expression. Additionally, I suggest authors discuss the molecular mechanisms in how AIF1/CASP3/CYCS/BCL2 protein levels are up regulated.
Alternatively, high expressions of the AIF1/CASP3/CYCS/BCL2 might be merely resulted from chemotherapy or radiotherapy sensitive type of cancer or not. In that case, authors can emphasize that two types of TNBC were identified.
Specific comments
- The names of genes should be typed in italic to avoid misunderstandings to be protein names. For example, in the first place, “AIFM1” (P1 L18) should be. Moreover, Gene ID (from NCBI Gene) can be indicated.
- P1-4, Introduction: This section is too long. It should be summarized within two pages at most. Some descriptions may be removed to the Discussion section.
- P3, Figure 1: The induction or activation of apoptosis-inducing/suppressing factors might have come from chemotherapies that patients have been taking. Authors can check if there are some correlations between specific anticancer drugs and the apoptosis-inducing effects. It would be important to indicate in this Figure if apoptosis-inducers/suppressors were up- or down-regulated by some specific drugs. I therefore recommend authors indicate which apoptosis-inducer/suppressor is targeted by a specific compound.
- P5, Figure 2, P6, Figure 3: Authors showed the results of statistical and immune-histochemical analyses to indicate that over-expression of AIF1 correlates with OS of TNBC patients. Similar results were shown by the analyses of CASP3. Readers will wonder why AIF1 protein accumulation correlates with CASP3 level. Authors had better discuss or describe more in detail.
- P5, Figure 2: Put scale bars in EFG as Figure 3G. That will clearly show that the magnification is the same without any explanations. Some panels in Figures 3, 4, 5, and 6 are missing that.
- P5, Figure 2: Legend to Figure 2H is missing. It is important. Because similar analyses were done in Figures 3 to 6.
- P9, Table 1: All the value in Pearson’s and Spearman’s methods is completely the same. In this case, Table 1 can be edited.
Minor comments
P10, L328, P12 L390: in vitro and in vivo should be typed in italic.
P1 L19, P3 L102, P9 L272, P12 L422, P15 L495: in silico should be typed in italic.
Author Response
We are grateful to your recommendations. We thoroughly went through our text including the Results and Discussion, made corrections and some additions. We intended the highlight our most significant results, their controversies and connections to earlier published relevant literature.
The vast majority of our patient samples were treatment naive at diagnosis/sample collection only 9/103 received neoadjuvant chemotherapy, but survival data do reflect the effect of therapies. Unfortunately, we have follow up biopsy after treatment from only very few patients, which does not allow to test for any statistical correlation.
In the 2nd paragraph of the discussion (it was relocated from the Introduction on your advice) we list some treatment including radio- and chemotherapies exploiting apoptosis as a mechanism for programmed cell death induction (see lines 279-291). We also cover this in the 5th paragraph of our discussion on AIFM1 (from line 316-319) and at the end of our 6th paragraph on CASP3 (lines 343-346) emphasizing the potential role of radiation therapy too, since most patients received chemo-/radiotherapy. However, even the “standard” chemotherapy, either with or without radiotherapy and the duration of treatments were diverse, that prevented us from making statistical conclusions which may link specific treatment and biomarker expression.
Unfortunately, in our specific cases we do not have exact data on how apoptosis markers are specifically upregulated in therapy naïve cases. Therefore, we would avoid any speculations on this, particularly considering the high molecular heterogeneity of TNBC, which may not allow to simplify this subgroup into only 2 types.
- Gene names such as “AIFM1” were put into italic.
- To shorten the Introduction its 5th paragraph was relocated to be the 2nd paragraph of the discussion. The last concluding sentence is also deleted from the Introduction.
- We fully agree that it would be useful to find direct relationships with treatment, such as using the BCL2 inhibitor venetoclax in the case of BCL2 expression. We refer to this with a sentence in lines 285-286 in our revised manuscript. However, please note that as mentioned above our patient samples used for immunohistochemistry were therapy naïve, therefore, we can’t generate such data in Figure 1.
- It is a good idea, thank you! We formed kappa correlation analyzes which correlates expression levels of different biomarker in individual cases, but only fair kappa correlations were only found between AIF1 and CASP3, AIF1 and CYCS, or between BCL2 and CASP3 found significant in Table2. Nevertheless, we added 1one sentence on this both into the results (lines 255-257) and to the end of the discussion (lines 431-433).
- We prefer this scale bar system, which is widely accepted. Instead of labeling each components separately, one scale bar can clearly show the exact magnifications for all adjacent components if clarified in the legends as we did.
- The legend for Fig. 2H is the last sentence of this legend but missing the letter (H) at the end which is now added.
- We replaced Table 1 for one, which contain both the Pearson and the Spearman’s correlations.
Both in silico and in vitro are changed to italic, but as far as I remember, this journal changed it back to normal typography in our earlier publications. We will see.
2 new references were included. One for radiotherapy and TNBC (1) and another on BCL2 inhibitor treatment (2)
- Chen H, Han Z, Luo Q, Wang Y, Li Q, Zhou L, Zuo H. Radiotherapy modulates tumor cell fate decisions: a review. Radiat Oncol. 2022 Dec 1;17(1):196. doi: 10.1186/s13014-022-02171-7.
- Alhoshani A, Alatawi FO, Al-Anazi FE, Attafi IM, Zeidan A, Agouni A, El Gamal HM, Shamoon LS, Khalaf S, Korashy HM. BCL-2 Inhibitor Venetoclax Induces Autophagy-Associated Cell Death, Cell Cycle Arrest, and Apoptosis in Human Breast Cancer Cells. Onco Targets Ther. 2020 Dec 31;13:13357-13370. doi: 10.2147/OTT.S281519.
Reviewer 3 Report
Comments and Suggestions for Authors
The Authors demonstrated that testing the expression of AIF1, caspase-3 and BCL2 in TNBC may identify a patient subgroup, which potentially benefit from tailored apoptosis-targeting treatment strategies.
Major comment: The selection of genes to be investiagted is not sufficiently justified.
English must be improved for clarity.
Consistent nomenclature is recommended, e.g., BCL2 /Bcl2 etc.
Figure 1 is not necessary as it does not add much novel data, and the experimental setup of the study is not complex.
Comments on the Quality of English Languagemust be comprehensively and extensively improved
Author Response
Many thanks for your recommendation and criticisms!
We obviously agree that the apoptosis pathway is more complex, but due to our immunohistochemistry approach, which is available for most pathology labs, we focused on some of its major markers including executioner elements. Fig. 1 as well as 4th paragraph of our introduction and the 2nd paragraph of our discussion may give a focused explanation both for the links among these markers and their relevance in apoptosis testing.
The nomenclature is made consistent, the gene symbols are consistently written in italic.
We prefer to keep our figure 1. which briefly summarizes the link among the tested markers and put in the simplified context of apoptosis (in line with your 1st point) and give a brief overview of our study design for easy visualization at the beginning of the manuscript (the Math&Meth part which could clarify this, is at the end of the manuscript).
Since we realized after submission that in the uploaded manuscript some needless hyphens remained through the copying process. Also, some mistyping, typographical errors and fragments were found from the earlier version in our manuscript. All these are corrected, English is improved, few expressions were replaced and sentences were shortened or were broken into two to support easier comprehension of our results and their interpretation. Our data included of multifaceted analyzes, which may not be easy to follow, but important details had to be presented.
Round 2
Reviewer 1 Report
Comments and Suggestions for Authors
The revised manuscript explores the prognostic relevance of apoptosis-related markers in triple-negative breast cancer (TNBC) by integrating in silico transcriptomic data and immunohistochemical (IHC) analysis of a retrospective clinical cohort. While the manuscript has been improved in several aspects, significant concerns remain:
1. Statements suggesting that AIF1, cleaved CASP3, and BCL2 expression may guide tailored apoptosis-targeting therapies are not adequately supported by functional or mechanistic data. The findings are correlative and hypothesis-generating at best. The text should be revised to reflect this limitation and avoid implying clinical utility prematurely.
2. Despite prior criticism, the study still relies solely on a single-center, retrospective cohort without validation in an independent dataset. This raises concerns about the robustness and generalizability of the reported findings. Incorporating external validation or acknowledging this limitation more explicitly is necessary.
3. Several markers (e.g., CASP8 and CYCS) show discordant prognostic associations between transcript and protein levels. These inconsistencies are described but not adequately contextualized. A deeper discussion on potential biological, technical, or post-transcriptional explanations is warranted.
4. The revision includes useful demographic and treatment details, but still lacks important clarifications:
a. Were treatment regimens (e.g., anthracyclines, taxanes) homogeneous across patients?
b. Were pre-treatment biopsies used in all cases?
c. How was treatment response defined and incorporated into analysis?
5. The immunoscoring approach is reasonable but remains subject to semi-quantitative bias. While interrater agreement statistics are presented, the modest kappa values between marker pairs suggest weak biological correlation, challenging the functional interpretations offered in the Discussion.
6. While the Discussion is expanded, it remains overly speculative in several areas and insufficiently critical regarding limitations such as:
a. The limited sample size
b. The absence of mechanistic or functional studies
d. The context-dependent prognostic role of BCL2 and CASP3, which are known to be inconsistent across TNBC subtypes
Overall, the manuscript has merit and presents clinically relevant data, but remains observational and exploratory. The conclusions must be appropriately tempered, and the discussion should better integrate conflicting literature and acknowledge key limitations.
Comments on the Quality of English LanguageThe English language throughout the manuscript is generally understandable but requires moderate revision.
Author Response
We are grateful for your further comments and criticism, which may help improve the clarity and quality of our paper besides acknowledging its limitations.
Please find our revisions (highlighted in red in the R2 manucript) and comments point-by-point below:
- Statements suggesting that AIF1, cleaved CASP3, and BCL2 expression may guide tailored apoptosis-targeting therapies are not adequately supported by functional or mechanistic data. The findings are correlative and hypothesis-generating at best. The text should be revised to reflect this limitation and avoid implying clinical utility prematurely.
- We modified our R1 conclusion statements according to your comment
Abstract: „Our results suggest that testing for the expression of AIF1, caspase 3 and BCL2 may identify a TNBC subgroup with favorable prognosis, warranting further research into the potential relevance of apoptosis-targeting treatment strategies.”
Discussion 1st paragraph: “Our findings may help identify a subgroup of TNBC patients with favorable prognosis, and offer rationale for future studies evaluating the potential of apoptosis-targeting strategies.”
Discussion, caspase 3 section: “This underscores the potential utility of immunohistochemistry in assessing apoptosis-related biomarkers, though further validation is required to determine its clinical value.”
“ Nevertheless, our results suggest that in TNBC, both AIF1 and the executioner cCASP3 in stock may contribute to the programmed cell death response following adjuvant chemo-/radiotherapy to support better patient outcome. However, the confirmation of this needs further functional studies.”
Conclusion: “Accordingly, testing for AIF1, cleaved CASP3 and BCL2 expression e.g. using immunohistochemistry may identify a TNBC subgroup with improved survival, although the clinical utility of apoptosis-targeting strategies remains to be established in futher studies.”
- Despite prior criticism, the study still relies solely on a single-center, retrospective cohort without validation in an independent dataset. This raises concerns about the robustness and generalizability of the reported findings. Incorporating external validation or acknowledging this limitation more explicitly is necessary.
We supplemented the last part of our conclusion accordingly: “ Despite our promising results, we acknowledge several limitations of our study, including its single-center and retrospective design, the absence of an independent external validation cohort, the lack of functional validation, and the moderate sample size. These may limit the generalizability and robustness of our findings. Therefore, future studies involving larger, multi-institutional cohorts and mechanistic investigations are required to endorse our observations.”
- 3. Several markers (e.g., CASP8 and CYCS) show discordant prognostic associations between transcript and protein levels. These inconsistencies are described but not adequately contextualized. A deeper discussion on potential biological, technical, or post-transcriptional explanations is warranted.
The potential reasons for discordant transcript and protein levels are now emphasized in a separate parapraph supported by new references.
„In this study, we observed differing prognostic strength of some biomarkers in the same TNBC cohort at different disease end-points (OS, RFS or DMFS). This did not raise major concerns if the same platform was used and could be explained e.g. by differences in treatment dynamics and duration up to these end-points. However, oc-casionally mRNA and protein abundance showed non-linear or even discordant prognostic correlations, experienced also by others in complex samples [57,58]. In general, technological, post-trancriptional and post-tanslational modifications can contribute to the discrepant mRNA and protein levels observed e.g. in our testing the prognostic role of CASP8 and CYCS expression [59,60]. The translation efficiency of mRNA abundance mainly depends on its stability/speed of decay besides microRNA regulation, which latter can silence mRNA translation [61]. On the other hand, post-translational mechanisms can regulate protein function and detectability, in-cluding, e.g. cleavage, phosphorylation, ubiquitination, acetylation or methylation [60]. Furthermore, method-related technical factors such as antibody sensitivity and fixation-related antigen preservation can also influence the efficiency of in situ protein detection with immunohistochemistry [62]. All these may contribute to the occasional inconsistent results of biomarker studies. Further standardization of pre-analytics and detection methods, as well as molecular studies on target gene transcription and translation are still needed to clarify the role of these modifying factors.”
- The revision includes useful demographic and treatment details, but still lacks important clarifications:
a. Were treatment regimens (e.g., anthracyclines, taxanes) homogeneous across patients?
b. Were pre-treatment biopsies used in all cases?
c. How was treatment response defined and incorporated into analysis?
A new paragraph was added on this:
„It is important to note that most of our clinical TNBC patients were therapy naïve at surgery/sample collection. Only 9 received diverse neoadjuvant therapies, which, however, resulted in no noticable pathological tumor regression by imaging follow up. Therefore, they were also included into this prognostic study and received similar therapy to the rest of 94 TNBC cases. The chemotherapy regimens administered to our TNBC cohort were heterogeneous, including various combinations of anthracyclines, taxanes, platinum agents, and other drugs (Anthracycline-based: 73 patients, Tax-ane-containing: 65 patients, Platinum-based: 3 patients, 5FU-containing: 16 patients, Avastin-containing: 1 patient), with or without radiotherapy (Table 4). While this re-flects real-world clinical practice, also introduces treatment-related variability that can influence survival outcomes. Unfortunately, our sample size relative to the diversity of treatment protocols did not allow stratified analyses by specific treatment type. This need to be tested in the future in large multicenter TNBC cohorts.”
- The immunoscoring approach is reasonable but remains subject to semiquantitative bias. While interrater agreement statistics are presented, the modest kappa values between marker pairs suggest weak biological correlation, challenging the functional interpretations offered in the Discussion.
„However, the only fair kappa correlations found in individual cases (kappa= 0.205-0.338) revealed their modest direct link in individual samples. On the other hand, some signal alternative pathways, which may partly interact and also contribute to the manifestation of their functions [18]. For example, the highest, almost good correlation was found between AIF1 and CYCS (correlates in 69 vs 34 patients), two cell respiratory oxidoreductases. They are released after mitochondrial outer membrane damage upon cell stress, and activate mainly alternative signaling. AIFM1 and cCASP3 (correlates in 64 vs 39 patients) may function at the executioner end-points of apoptosis through partly alternative pathways [33]. BCL2 inhibits mitochondrial membrane damage induced by pro-apoptotic factors that may activate CASP3 (correlates in 65 vs 38 patients). Despite opposing functions, even their modest positive prognostic association reflects their complex regulation. However, the molecular heterogeneity of TNBC subtypes call our attention to the context- and cohort dependence of all prognostic biomarker studies [6].”
We supplemented the end of the 1st section of the Math&Meth as:
„It is important to note, that most TNBC samples were therapy naïve, only 9 cases re-ceived neoadjuvant chemotherapy, but without recognizable regression by imaging follow up. Therefore, all 103 patients received adjuvant therapy and their prognostic data were considered accordingly.
- While the Discussion is expanded, it remains overly speculative in several areas and insufficiently critical regarding limitations such as:
a. The limited sample size
b. The absence of mechanistic or functional studies
d. The context-dependent prognostic role of BCL2 and CASP3, which are known to be inconsistent across TNBC subtypes
As you recommended, we discuss these points with more details in the R2 manuscipt. Our responses to your point 1 and 2, our new paragraphs at the end of the discussion and some additional short statements also in this chaper (all hghlighted in red), reflect on these points.
Novel references added to our R2 manuscript:
Maier T, Güell M, Serrano L. Correlation of mRNA and protein in complex biological samples. FEBS Lett. 2009 Dec 17;583(24):3966-73. doi: 10.1016/j.febslet.2009.10.036. PMID: 19850042.
Schaefke B, Sun W, Li YS, Fang L, Chen W. The evolution of posttranscriptional regulation. Wiley Interdiscip Rev RNA. 2018 Sep;9(5):e1485. doi: 10.1002/wrna.1485.
Zhong Q, Xiao X, Qiu Y, Xu Z, Chen C, Chong B, Zhao X, Hai S, Li S, An Z, Dai L. Protein posttranslational modifications in health and diseases: Functions, regulatory mechanisms, and therapeutic implications. MedComm (2020). 2023 May 2;4(3):e261. doi: 10.1002/mco2.261.
Oliveto S, Mancino M, Manfrini N, Biffo S. Role of microRNAs in translation regulation and cancer. World J Biol Chem. 2017 Feb 26;8(1):45-56. doi: 10.4331/wjbc.v8.i1.45.
Lin, F., Shi, J. (2022). Standardization of Diagnostic Immunohistochemistry. In: Lin, F., Prichard, J.W., Liu, H., Wilkerson, M.L. (eds) Handbook of Practical Immunohistochemistry. Springer, Cham. https://doi.org/10.1007/978-3-030-83328-2_2
Reviewer 3 Report
Comments and Suggestions for Authors
The Authors have addressed all comments appropriately.
Author Response
Thank you very much for accepting the revisions and improved English made in our R1 manuscript!
Please note and see, that we have made some more changes on the R1 manuscrip in R2 based on the recommendations of reviwer 1.
Round 3
Reviewer 1 Report
Comments and Suggestions for Authors
The revised manuscript presents a substantial improvement over the previous version and addresses nearly all major concerns raised during the last round of review. The authors have made commendable efforts to clarify the mechanistic interpretations, contextualize mRNA–protein discrepancies, and acknowledge the retrospective and single-center limitations of their study.
Notably, the discussion now appropriately reflects the correlative and exploratory nature of the findings, avoiding overstatements about clinical applicability. The inclusion of detailed methodological clarifications regarding patient treatments, biopsy timing, and immunohistochemical scoring enhances the transparency and reproducibility of the work. The newly added discussion on biological and technical explanations for discordant transcript–protein relationships is well-referenced.
That said, a few minor issues remain:
1. Some sentences in the discussion remain overly long or complex. A light professional language edit would improve clarity and readability.
2. While the discussion acknowledges weak inter-marker correlations, a more neutral tone in interpreting modest kappa values would better align with the data.
3. The figures are improved, but further refinements (e.g., font size and consistency across panels) could enhance visual clarity in the final production stage.
Overall, the manuscript now presents a valuable and well-balanced contribution to the understanding of apoptosis-related biomarkers in the prognosis of TNBC. With minor editorial polishing, it is suitable for publication.
Comments on the Quality of English LanguageThe English language in the revised manuscript is generally clear and comprehensible. However, several sections, particularly in the Discussion, contain overly long or complex sentences that could benefit from restructuring for clarity. Occasional grammatical issues, awkward phrasing, and inconsistent punctuation are also present.
Author Response
Thank you again for calling our attention to further improvement and correction possibilities!
You can find all of our changes and corrections highlighted either in red or through the correction signs of MSWord.
- Some sentences in the discussion remain overly long or complex. A light professional language edit would improve clarity and readability.
Some sentences were broken into two, shortened or further clarified according to your suggestions (see examples below and in R3 manuscript). However, the multiplex and complex topic make it difficult to simplify significantly further without loss of important details and affecting the fidelity e.g. of the referred literatures presented.
Abstract – conclusion:
Revised: Our results suggest that testing for the expression of AIF1, caspase 3 and BCL2 may identify partly overlapping TNBC subgroups with favorable prognosis, warranting further research into the potential relevance of apoptosis-targeting treatment strategies.
End of 1st paragraph in the Discussion:
Revised: Apoptosis can be induced e.g. by treatment-related cell stress caused by radiotherapy [22] or chemotherapy (e.g. using doxorubicin, cisplatin, or 5-fluorouracil), which cause DNA damage and mitochondrial dysfunction [23]. Our results show that testing for the expression of BCL2 and the executioner apoptotic factors AIF1 (gene AIFM1) and caspase 3 (gene CASP3) may identify partly overlapping TNBC subgroups with favorable prognosis. These findings may offer rationale for future studies evaluating the potential of apoptosis-targeting treatment strategies in this aggressive tumor.
Discussion:
lines 315-318.
Original: The apoptosis inducing factor-1 encoded by the AIFM1 gene is an oxidoreductase flavoprotein of the mitochondrial respiratory complexes, which upon release can also be an important effector of caspase independent apoptosis though triggering endonucleases for a controlled DNA cleavage [30-33].
Revised: The apoptosis inducing factor-1 encoded by the AIFM1 gene is an oxidoreductase flavoprotein of the mitochondrial respiratory complexes. When released from mitochondria, it can trigger endonuclease activation and controlled DNA cleavage, acting as a mediator of caspase-independent apoptosis [30-33].
Lines 326-329.
Original: This reverse-phase protein array (RPPA) based test revealed massive survival benefits for patients of elevated AIFM1 protein expression not only for OS but also for RFS and DMSF, even at the most stringent conditions where high and low expression patient groups were separated at the median marker level.
Revised: This reverse-phase protein array (RPPA) based test revealed massive survival benefits for patients of elevated AIFM1 protein expression not only for OS but also for RFS and DMSF. These associations remained significant even when using strict median-based thresholds to separate high and low expression groups.
- While the discussion acknowledges weak inter-marker correlations, a more neutral tone in interpreting modest kappa values would better align with the data.
Please note our concluding statements from the Discussion above (at point 1.) where we emphasize that the marker selected subgroups are only partly overlap at the patient level. This is also clarified in the concluding paragraph of our Discussion:
The biomarker levels only moderately overlapped in individual patient samples, which underline the complex regulation of apoptotic pathays. …Accordingly, testing for AIF1, cleaved CASP3 and BCL2 expression e.g. using immunohistochemistry may identify partly overlapping TNBC subgroups with improved survival. However, the clinical utility of apoptosis-targeting strategies remains to be established in further studies….
- The figures are improved, but further refinements (e.g., font size and consistency across panels) could enhance visual clarity in the final production stage.
We corrected all of our figures concerning positioning and font harmonization as much as it was possible. Please note that our graphs were created by different software packages and e.g. K-M Plotter did not allow free positioning of data and font changes. Nevertheless, we did what was possible to improve their visual harmony and quality.
Overall, the manuscript now presents a valuable and well-balanced contribution to the understanding of apoptosis-related biomarkers in the prognosis of TNBC. With minor editorial polishing, it is suitable for publication.
We went through thoroughly again on the English grammar, punctuation and typographical errors and corrected them.